# Dietary diversity and its correlates among pregnant adolescent girls in Ghana

**Linda Afriyie Gyimah**[ID][1]*, **Reginald Adjetey Annan**[1], **Charles Apprey**[1], **Anthony Edusei**[2], **Linda Nana Esi Aduku**[1], **Odeafo Asamoah-Boakye**[ID][1], **Wisdom Azanu**[3], **Herman Lutterodt**[4]

**1** Faculty of Biosciences, Department of Biochemistry and Biotechnology, College of Science, Kwame Nkrumah University of Science and Technology, Kumasi, Ghana, **2** Department of Community Health, School of Public Health, Kwame Nkrumah University of Science and Technology, Kumasi, Ghana, **3** Department of Obstetrics and Gynecology, University of Allied Health Sciences, Ho, Ghana, **4** Department of Food Science and Technology, Kwame Nkrumah University of Science and Technology, Kumasi, Ghana

* lindagyimah17@gmail.com

## Abstract

### Background

Dietary diversity, a qualitative measure of dietary intake, which reflects the variety of foods consumed has been recommended to assuage nutritional problems related to insufficient micronutrients, and food insecurity. To better understand the underlying factors for poor birth outcomes in Ghana, we assessed factors associated with dietary diversity among rural and urban pregnant adolescents in the Ashanti Region of Ghana.

### Methods

As part of a larger longitudinal cohort of 416 pregnant adolescents, the FAO minimum dietary diversity for women index was used to determine the dietary diversity score (DDS) of the participants from a previous days' 24-hour dietary recall data. The household hunger scale (HHS) and lived poverty index (LPI) were used to determine hunger and socioeconomic status. Eating behavior and socio-demographic data were gathered using interviewer-administered questionnaires.

### Results

The mean age of the participants was 17.5 (±1.4) years with an MDD-W of 4.4 and 56% recording inadequate MDD score. More rural (63.6%) than urban dwellers (50.6%) had inadequate DDS (p = 0.008). Among all the multiple variables tests of associations on dietary diversity, only hunger status (p = 0.028) and both food aversion and poverty status (p = 0.003) had a significant effect on the adolescents' dietary diversity. Rural dwelling adolescents (AOR = 1.7, p = 0.035, 95% CI = 1.0–2.6) recorded higher odds for inadequate DD compared with the urban respondents. Pregnant adolescents with severe hunger had higher odds (Unadjusted OR = 1.9, p = 0.053, 95% CI 1.1–3.8) for inadequate dietary diversity compared with those with no hunger.

**Data Availability Statement:** Data cannot be shared publicly because it contains sensitive identifying information. However, data are available

from the Committee on Human Research Publication and Ethics (CHRPE), the ethics board of the School of Medical Sciences of the Kwame Nkrumah University of Science and technology, KNUST and Komfo Anokye Teaching Hospital (KATH), Kumasi, Ghana (Email: chrpe.knust.kath@gmail.com) for researchers who meet the criteria for access to confidential data.

**Funding:** Funding for this study was provided by the Nestle Foundation. The funders had no role in study design, data collection and analysis, decision to publish, or preparation of the manuscript.

**Competing interests:** The authors have declared that no competing interests exist.

## Conclusions

Inadequate DD is common among pregnant adolescents in this study and is associated with rural living, food insecurity, poverty, and food craving. Livelihood support for pregnant teenagers and nutrition education are recommended interventions to improve dietary quality and limit the consequences of poor dietary diversity.

## 1. Introduction

Pregnancy poses a significant risk for malnutrition [1] because extra nutrients are needed to support the growth of the developing foetus. For the adolescent, this risk is heightened due to the continuous psychological, social, and physical development of the pregnant girl [2]. Food insecurity and malnutrition continue to be a challenge worldwide and developing countries are the most affected [3]. In developing countries, micronutrient deficiencies are predominant among women of reproductive age [4], with numerous unfavorable effects on women's health, work, and survival. To lessen the causes of malnutrition globally, the first two goals of the Sustainable Development Goals (SDGs) seek to reduce poverty and end all forms of malnutrition and hunger by 2030 [5]. One major contributor to maternal malnutrition in developing countries according to a systemic review of 62 studies carried out in developing countries is inadequate dietary intake mainly due to inadequate quantity, quality and diversified diets, poverty, high-intensity agricultural labor, and high fertility rate [6]. Accordingly, findings revealed that inadequate intakes of macronutrients and micronutrients in pregnant women are common due to their typically monotonous diets, based largely on plants or cereals and deficient of micronutrient dense foods such as vegetables, fruits and animal sources [6]. Dietary diversity is one of the recommended strategies to improve maternal diet and prevent malnutrition, through increased daily intake of the various food groups in one's diet [7].

Dietary diversity (DD) has also been used as a representation for nutrient adequacy and quality of dietary intake and particularly, the probability of micronutrient adequacy in a diet [8]. It is a qualitative assessment of food intake that reflects an individual's access to a variety of foods. The Minimum Dietary Diversity for Women (MDD-W) is a dichotomous predictor of whether or not women 15–49 years of age have consumed at least five out of ten specified food groups the previous day or night. The proportion of women between 15–49 years of age who meet this minimum in a population can be used as a proxy for higher micronutrient adequacy, a significant dimension of diet quality [9]. Dietary Diversity has been linked with socio-demographic factors among pregnant women, including education level, employment status, monthly income, household assets, and land ownership [10,11] and food security [12]. For example, pregnant Ethiopian women with secondary or tertiary education, higher household income, land ownership, and receiving emotional support from husbands had better DD [7]. [13] however observed no association between DD and socio-demographic, or socioeconomic status in pregnant women.

Household hunger scale (HHS) is used to assess food deprivation in a population and together with other tools like DD and income, HHS can be used to measure some aspects of food security [14]. Lived poverty is an index that measures the frequency with which people experience shortages of basic necessities during the year [15]. LPI measures a portion of the concept of poverty that is not well assessed by other measures. Several authors have investigated that people living in poverty are not exposed to resources which improve health, such as

food market, a supply of nutritious food, water quality, recreational facilities, housing conditions, employment, education, and access to medical care [16–18].

During pregnancy, eating behaviors change. Food cravings and aversions are indicators of physiological stress during pregnancy [19,20] as a result of changes in metabolic states and hormone levels [21]. Pregnant women usually crave sweet, salty, fatty, or spicy foods which are generally considered unhealthy [22] and avert certain foods that can limit the intake of a variety of dishes. Studies on food craving and aversions indicate that pregnant women usually crave energy-dense foods [23] and avert nutritious diets such as cereals, meat, fish [19,24] and fruits like mangoes and pineapples [25]. Pica practice is a form of craving which involves the ingestion of non-food substances like clay, soap, ashes, paint, paper among others. Pica has been of public interest due to its high prevalence in pregnant women. It is useful for defense against pathogens and toxins, quelling nausea, vomiting, and diarrhea. [26]. It may also be harmful as it reduces the bioavailability of beneficial nutrients and introducing toxic substances [26].

In Ghana, about 14.4% of adolescent girls aged between 15 and 19 years have begun childbearing and about 12% have a live birth [27]. Adolescent pregnancy in Ghana has been linked to socio-economic status, ignorance, and their inability to access information and services related to reproductive health [28], but dietary diversity and factors associated with it among pregnant adolescents in Ghana have not been studied. Studies conducted in Bangladesh [29] and Malawi [30] reported 5.2 and 4.0 as the MDD of pregnant adolescents respectively. Whilst 70% of Malawian pregnant adolescents did not meet the MDD, only 33.3% of Bangladesh pregnant adolescents did not consume from more than five food groups. Since a diverse diet indicates a healthier diet, good DD can promote good pregnancy outcomes. This has been shown in studies [31,32] conducted on pregnant women in Ghana and Ethiopia where participants with inadequate dietary diversity score presented an increased risk of presenting adverse birth outcomes. The study in Ghana presented a significantly higher proportion of LBW in women with low DDS during pregnancy (60.7%) compared to women who had high DDS during pregnancy (39.6%). LBW (15.6%), preterms (22.6%), and stillbirths (8.1) were greater in women with inadequate diets than those with adequate diets (2.7%, 4.8%, and 1.1%). Again, women with inadequate diets had a higher risk of maternal anemia (ARR: 2.29; 95% CI: 1.62, 3.24) than women in the adequate group ($P < 0.05$). Therefore, improving DD is an important step for ensuring daily intake of the various food groups, ensuring adequate maternal diets, preventing malnutrition [7], and promoting desirable birth outcomes. Particularly, understanding the correlates of DD among pregnant women in Ghana is a prerequisite for addressing any shortfalls in the Ghanaian context. This study assessed the dietary diversity and its associated factors among pregnant adolescents in Ashanti Region, Ghana.

## 2. Method

### Study setting

The Ashanti Region is located in the southern sector of Ghana occupying about 10% of the total land area of Ghana. In terms of population, it is the most populated region in Ghana, with an estimated population of 5,792,187, accounting for 19% of Ghana's total population [33]. The region is divided into 31 districts with Kumasi Metropolis being the most populated and the second-largest city in Ghana. Each district has a district hospital, health centers and several Community-based Health Planning Services (CHPS) compounds depending on the size of the district [34]. Due to its position in the middle belt of Ghana, the Ashanti region was chosen for this study and also because the majority of maternal and child health studies have been conducted in both the Northern and Southern sectors of Ghana [31,35]. A rural area is a

community with less than 5000 population while all others are considered urban [34]. By this classification, the Ashanti region is about 61% urban.

## Study design

This study is part of a larger longitudinal study known as the Adolescent Nutrition Birth Outcomes, Ghana (ANBOG) study. The study sought to assess the underlying factors associated with poor nutrition and anemia and how they predicted birth outcomes among pregnant adolescents, aged 13–19 years in the Ashanti Region of Ghana. It involved 3 phases; baseline, ethnographic, and follow-up. Some aspects of the baseline phase which involved socio-demographics and dietary intake were used for this study. Low Birth Weight was the main outcome of interest but other birth outcomes such as preterm, stillbirths, and hemorrhagic births was also collected. We proposed that poorly nourished adolescents would have a larger average proportion of LBW compared with the better-nourished ones. These pregnant adolescents, recruited at an average gestational age of 16.0 weeks from May to December 2018 were followed up until they delivered. Follow up data on birth outcomes were collected from January to July 2019.

## Selection of health facilities

The pregnant adolescents were recruited from Kumasi Metropolis and selected Districts in the Ashanti Region of Ghana. The initial intention was to have a good number of pregnant adolescents from both urban and rural areas of the Ashanti Region. Two rural districts Ahafo Ano South and Asanti Akyem South were initially conveniently selected and all health centers and CHPS compounds were eligible for the recruitment of adolescent pregnant girls. However, it was difficult to get them attending the antenatal services, where recruitment occurred, even with special announcements made in their respective communities. These led to visits to other 3 rural districts: Bosumtwi, Asante Akim North, and Ahafo Ano North districts. For the urban district, Kumasi Metropolis (the regional capital) was chosen initially, but Asante Akim Central and Ejisu Juaben were later added to reach the required sample size.

## Sample size calculation

A sample size of 460 was statistically determined using a formula based on sample size based proportion [36]:

$$n = 2(Z\alpha/2 \ + \ Z\beta)2\,p(1-p)/(P1-P2)^2.$$

Where, n = sample size, Z α/2 = 1.96 at type 1 error of 5%, Z β = 0.84 at 80% power, P1 = LBW in pregnant adolescents with adequate nutritional status, P2 = LBW in pregnant adolescents with poor nutritional status, p1-p2 = difference in prevalence of low birth weight between pregnant adolescents with adequate nutritional status at birth and those with inadequate nutritional status, and p = pooled prevalence = (p1 +p2)/2. In a pilot study among pregnant adolescents in the area [37], 23.3% had LBW children (that is birth weight less than 2.5 kg). We proposed that LBW in pregnant adolescents with adequate nutritional status would be 11.4%, while those with poor nutritional status would remain 23.3%, a reduction of just above half (51.1%). Hence, p1 = 11.4%, p2 = 23.3%, their proportions being p1 = 0.114 and p2 = 0.233, and p = (0.114 +0.233)/2 = 0.1735.

Using the above descriptives, the sample size n = 2(1.96+0.84) $^2$ x 0.1735(1–0.233)/(0.114–0.233) $^2$, n = 2.09/0.01, equal 209 was calculated, which implied we needed to recruit 209 participants in each arm of the study (half in the poorly nourished group and a half in the well-

nourished group) making 418 participants showing a significant association between poor nutrition and LBW. However, we added 10% attrition to give 460 participants who were needed.

## Recruitment

The study was conducted in the hospital/health center/CHPS compounds-based. Four hundred and sixteen (416) pregnant adolescents were recruited within the period of the baseline data collection. Participants were recruited from hospitals and health centers during maternity care visits. Most of these hospitals have maternity clinics days for only pregnant adolescents. On these dates, researchers visited the facilities, and any pregnant adolescent within the required age group who gave consent was recruited for the study. For health centers in rural areas, announcements were made at community information centers to invite pregnant adolescents to the health centers on specific dates. These announcements were necessary as the adolescents were not visiting the health centers for fear of stigma. There was no randomization in selecting the districts and in selecting the girls at the health centers because of the few numbers that attended antenatal services. The selection of the girls was on a first-come-first-serve basis until the sample size was reached. Yet, it took 6 months to reach the sample size.

## Data collection

Full details of the longitudinal study are outside the scope of this paper and will be described elsewhere. The baseline of the larger study was used for this study. For this paper, a standardized questionnaire was used to collect data on socio-demographic descriptives of pregnant adolescents. Data on age and parity were verified from their National Health Insurance Identification cards and maternal health record books, and dietary intake was assessed to determine DD. Data collection was done by postgraduate nutrition and dietetics students and other enumerators with at least tertiary degrees/diplomas. All data collection participants were given 2 days of training before data collection, including a trial in a hospital that was not included in the actual study. The data collection teams were paired in a way that enumeration using the questionnaire was done by the non-nutritionists, while the dietary assessment was performed by the trainee nutritionists.

**Assessment of dietary intake and eating behaviour.** Dietary intake was assessed using the repeated 24-hour dietary recall method which was taken three times (two weekdays and one weekend). The participants were asked to recall all food and beverages consumed for the previous day; two weekdays and a weekend. Household handy measures were used to aid participants in the estimation of portion sizes of their food intake and these portions were converted into grams and levels of nutrient intake using a nutrient analysis Microsoft excel software designed by the University of Ghana, Department of Food Science and Nutrition [38]. For this study, the first 24-hour dietary recall was used to determine dietary diversity.

**Independent variables: Socio-demographic, household hunger scale and lived poverty index**. Socio-demographic data collected on adolescents using the structured questionnaire included age, marital status, employment, income, levels of education, and parity. Marital status, education, and employment were collected as categorical data while age, income, and parity were scaled variables. The Household Hunger Scale [14] was used to determine the prevalence of hunger in the households these participants came from. The questionnaire constituted 3 questions. Question 1 asked if there was ever no food to eat of any kind in their houses because of a lack of resources to get food. If yes, the next question asked how often. The second question asked if any household member went to sleep at night hungry because there was not enough food and for those who answered yes, how often. The final question asked if

any household member went a whole day and night without eating anything at all because there was not enough food, and for those who answered yes, how often. The responses to these questions were coded and scored to provide scores ranging from 0–6. Scores < 2 meant little or no hunger, 2–3 meant moderate hunger and 4–6 meant severe hunger [14].

To determine the Lived Poverty Index (LPI) score, responses given by the respondents to the LPI questions about the availability of food, water, cash income, medical care, and cooking fuel over the past year were collected. For each question, respondents provided scores, ranging along a five-point scale from 0 (which can be thought of as no lived poverty) to 4 which would reflect a constant absence of all basic necessities (Mattes, 2008) [15]. The averages of the scores were then categorized into low (0–1.0), moderate (1.01–1.5), and high (> 1.5) LPI.

**Dependent variable: Dietary diversity**. The FAO's Minimum Dietary Diversity for Women (MDD-W) was used to determine maternal dietary diversity [9]. The Minimum Dietary Diversity for Women (MDD-W) is a population-level indicator of diet diversity validated for women aged 15–49 years old. The MDD-W is a dichotomous indicator based on 10 food groups, namely; grains, white roots and tubers, plantain; pulses (beans, pea, lentils); nuts and seeds; dairy; meat, poultry, and fish; eggs; dark green leafy vegetables; other vitamin-A rich fruits and vegetables; other vegetables; other fruits [9] and is considered the standard for measuring population-level dietary diversity in women of reproductive age. The previous days' 24-hour recall from each participant were used to calculate the MDD-W. All responses in the affirmative (Yes) are scored 1 whilst food groups that were not consumed (No) were scored 2. The food groups consumed (Yes = 1) are then summed into a score ranging from 0 to 10. A score of 5 or more (consumption of five or more food) was categorized into adequate [5–10] and a score of less than 5 (consumption of less than 5 food groups) inadequate (0–4). A questionnaire was used to obtain data on food cravings, pica practices, and food aversions.

## Ethical clearance

Ethical clearance for the study (Reference: CHPRE/ AP/236/18) was obtained from Committee on Human Research, Publications and Ethics of the Kwame Nkrumah University of Science and Technology, Kumasi, Ghana (CHRPE/KNUST), and personal or parental consent were obtained from participants before the beginning of the study.

## Data analysis

Data collected were first entered into Microsoft Excel 2019 and thoroughly cleaned to eliminate errors. Then, data was imported into Statistical Package for Social Sciences version 25 (SPSS IBM Inc Chicago, USA) for statistical analysis. Sociodemographic characteristics, eating behaviors, hunger, and poverty status was used as independent variables while minimum dietary diversity was used as the main outcome variable (dependent variable). Descriptive statistics were done and reported as relative frequencies for sociodemographic characteristics, eating behaviors, hunger and poverty status, and dietary diversity. A test of normality using the Kolmogorov-Smirnov test was done to ascertain whether the continuous variables were normally distributed. A chi-square (Fisher's exact test) cross-tabulation was performed to compare frequencies of sociodemographic characteristics, eating behaviors, hunger, and poverty status by community type and dietary diversity. Independent t-test and one-way ANOVA were used for parametric mean comparisons of the continuous variables. Non-parametric tests (Mann Whitney 'U' test and Kruskal Wallis test) was performed to compare mean differences between age, HHS, LPI, and DDS. A bivariate correlation was used to determine the association between age, HHS, LPI, and DDS. A binary logistic regression analysis was performed to determine predictors of dietary diversity, and these are reported as odds ratios, to explain the

combined effect size of the independent variables. All tests were 2-tailed, and differences were considered statistically significant at p < 0.05.

## 3. Results

### Socio-demographic characteristics

The sociodemographic characteristics of the participants are presented in Table 1. The majority of the participants were unmarried (76.0%), unemployed (71.6%), lived on no income (74.5%), had only completed basic education (61.3%), and were between the ages of 16 and 19 years (92.3%). More urban (25.1%) than rural (14.5%) pregnant adolescents completed secondary education (p = 0.018). Though not significant, a greater proportion of rural than urban participants were younger teenagers (8.1% versus 7.4%, p = 0.853) and were employed (31.2% versus 26.3%, p = 0.321).

### Food intake, dietary diversity, hunger, poverty

As indicated in Fig 1, the main food groups consumed the previous day were grains, roots and tubers (99.3%), meat, poultry and fish (86.3%), and other vegetables (94%). Less than 4 in 10

**Table 1. Distribution of socio-demographic characteristics of participants.**

| CHARACTERISTICS | MEAN±SD | TOTAL (%) | RURAL (%) | URBAN (%) | P-VALUE |
|---|---|---|---|---|---|
| Age group (Years) | 17.5 ± 1.4 | | | | 0.853[¥] |
| 13–15 | | 32 (7.7) | 14 (8.1) | 18 (7.4) | |
| 16–19 | | 384 (92.3) | 159 (91.9) | 225 (92.6) | |
| Total, n (%) | | 416 (100) | 173 (41.6) | 243 (58.4) | |
| Marital Status | | | | | 0.563[¥] |
| Single | | 316 (76.0) | 134 (77.5) | 182 (74.9) | |
| Married | | 100 (24.0) | 39 (22.5) | 61 (25.1) | |
| Occupation | | | | | 0.321[¥] |
| Unemployed | | 298 (71.6) | 119 (68.8) | 179 (73.7) | |
| Employed | | 118 (28.4) | 54 (31.2) | 64 (26.3) | |
| Parity | | | | | 0.485[¥] |
| 1 | | 316 (76.0) | 128 (74.0) | 188 (77.4) | |
| >1 | | 100 (24.0) | 45 (26.0) | 55 (22.6) | |
| Education | | | | | **0.018[‡]** |
| None | | 19 (4.6) | 8 (4.6) | 11 (4.5) | |
| Primary | | 56 (13.5) | 31 (17.9) | 25 (10.3) | |
| JHS | | 255 (61.3) | 109 (63) | 146 (60.1) | |
| SHS | | 86 (20.7) | 25 (14.5) | 61 (25.1) | |
| Income (₵) | | | | | 0.173[‡] |
| No Income | | 310 (74.5) | 135 (78.0) | 175 (72.0) | |
| Below 100 | | 41 (9.9) | 19 (11.0) | 22 (9.1) | |
| Between 100–500 | | 61 (14.7) | 18 (10.4) | 43 (17.7) | |
| More than 500 | | 4 (1) | 1 (0.6) | 3 (1.2) | |

Data are reported as frequency (percentage), JHS- Junior High School, SHS- Senior High School,

[‡] Chi-Square p-value, some cells were less than 5 for chi-square,

[¥] Fisher's exact test p-value.

bold values are significant at p < 0.05.

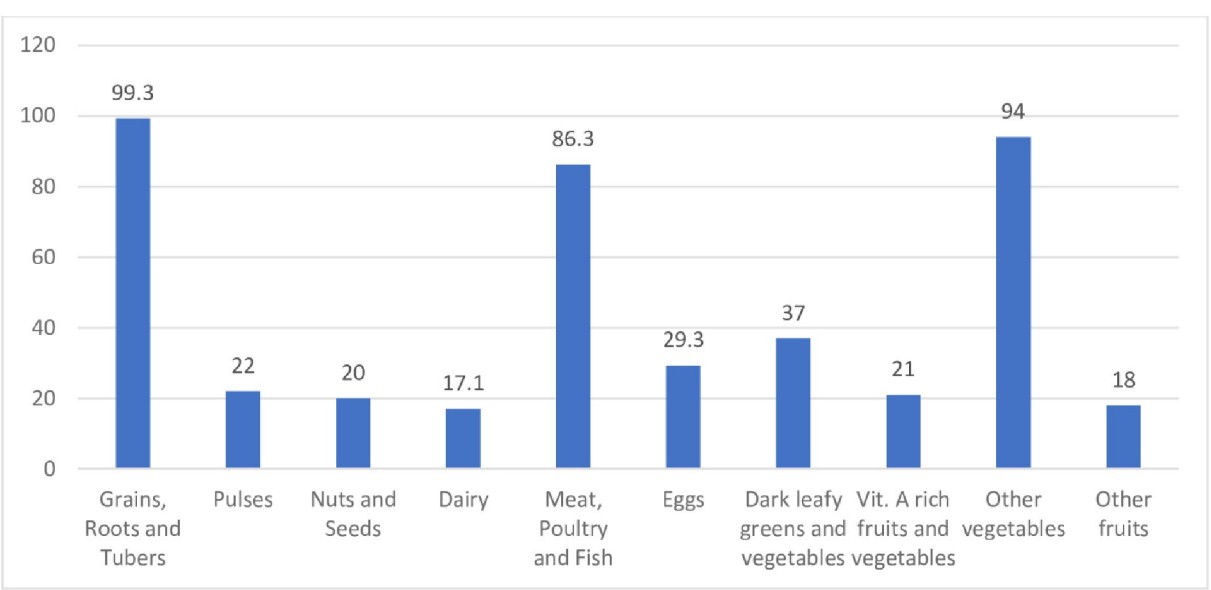

**Fig 1. Proportions of adolescents consuming different food groups previous the 24 hours.**

respondents consumed dark green leafy vegetables, and less than 3 in 10, consumed eggs (29.3), pulses (22%), nuts and seeds (20%), and vitamin A-rich fruits and vegetables (21%).

Table 2 presents the Dietary Diversity, Household Hunger, Lived Poverty Index, and eating behavior of participants. The mean MDD-W score was 4.4, however, more than half of the participants (56%) had inadequate dietary diversity, meaning they consumed less than 5 food groups the previous day. More than a fifth of the participants (27.1%) were food-deprived, while 36.5%, fell within the high poverty category. More than a third (38.0%) of the participants practiced pica, 64.2%, did crave for food and non-food substances, and 44.5%, had food aversion during pregnancy. A higher proportion of rural than urban participants was deprived of food (27.2% versus 1.2%, p<0.001) and lived in abject poverty (51.4% versus 25.9%, p<0.001). Although more rural participants practiced food craving, pica was practiced higher among urban dwellers than rural dwellers.

## Levels of deprivation among respondents

Fig 2 indicates the percentage of participants who went through pregnancy year without basic necessities. Overall, about 43% and 32% of the participants reported going without income and food at least once or twice in the past year respectively. Two in ten (23%) participants reported having shortages in medical care at least once or twice in the past year whilst approximately 7% of the participants went without water.

## Association between socio-demographics and MDD-W

Table 3 presents proportions of sociodemographic characteristics who had adequate and inadequate dietary diversity. A higher proportion of the rural (67.1%) than urban (53.9%) pregnant adolescents had inadequate dietary diversity (p = 0.008). Participants who had lower educational achievement (that is, completed primary school) presented the largest proportion (67%) of those with inadequate dietary diversity as well as, the least mean dietary diversity (4.2 ± 1.0) (p = 0.031). Dietary Diversity did not vary by age, marital status, parity, occupation, and

**Table 2. Dietary diversity, household hunger, lived poverty index and eating behaviour of participants.**

| VARIABLES | MEAN±SD | TOTAL (%) | RURAL (%) | URBAN (%) | P-VALUE |
|---|---|---|---|---|---|
| **MDD-W categories** | **4.4±1.30** | | | | **0.009¥** |
| Inadequate | | 233 (56.0) | 110 (63.6) | 123 (50.6) | |
| Adequate | | 183 (44.0) | 63 (36.4) | 120 (49.4) | |
| **Household Hunger Scale** | | | | | **<0.001‡** |
| No Hunger | | 303 (72.8) | 88 (50.9) | 215 (88.5) | |
| Moderate Hunger | | 63 (15.1) | 38 (22.0) | 25 (10.3) | |
| Severe Hunger | | 50 (12.0) | 47 (27.2) | 3 (1.2) | |
| **Lived Poverty Index** | | | | | **<0.001‡** |
| Low | | 194 (45.2) | 50 (30.1) | 142 (58.4) | |
| Moderate | | 75 (18.3) | 33 (19.1) | 42 (17.3) | |
| High | | 147 (35.3) | 88 (50.9) | 59 (24.3) | |
| **Food Aversion** | | | | | **0.046¥** |
| Yes | | 185 (44.5) | 87 (50.3) | 98 (40.3) | |
| No | | 231 (55.5) | 86 (49.7) | 145 (59.7) | |
| **Food Craving** | | | | | 0.078¥ |
| Yes | | 267 (64.2) | 120 (69.4) | 147 (60.5) | |
| No | | 149 (35.8) | 53 (30.6) | 96 (39.5) | |
| **PICA Practice** | | | | | **0.032¥** |
| Yes | | 160 (38.5) | 56 (32.4) | 104 (42.8) | |
| No | | 256 (61.5) | 117 (67.6) | 139 (57.2) | |

Categorical data are presented as frequency (percentage),

‡ Chi-Square p-value,

¥ Fisher's exact test p-value.

Bolded p-values are significant at p < 0.05. MDD-W: Minimum Dietary Diversity–Women determined from intake from 10 food groups the previous day.

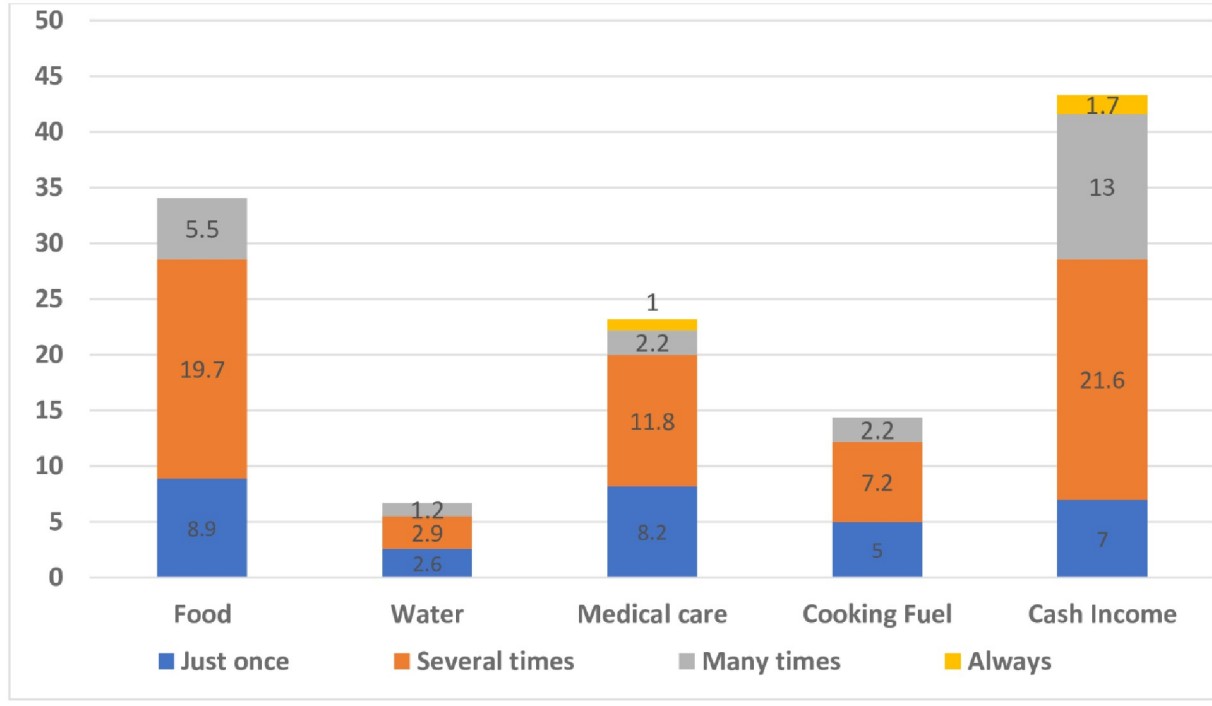

**Fig 2. Percentage of participants who went without basic necessities.**

**Table 3. Relationship between dietary diversity and socio-demographic characteristics.**

| VARIABLES | | | DIETARY DIVERSITY (DD) | | $X^2$, P VALUE |
|---|---|---|---|---|---|
| | MEAN±SD | P-value | INADEQUATE | ADEQUATE | |
| **Mean DD** | 4.4 ± 1.3 | | | | |
| Community Type | | | | | |
| Rural | 4.2 ± 1.2 | **0.008[†]** | 110 (63.6) | 63 (36.4) | **6.896, 0.009[¥]** |
| Urban | 4.6 ± 1.3 | | 123 (50.6) | 120 (49.4) | |
| **Age group (Years)** | | | | | |
| 13–15 | 4.9 ± 1.4 | 0.061[†] | 16 (50.0) | 16 (50.0) | 0.508, 0.579[¥] |
| 16–19 | 4.4 ± 1.3 | | 217 (56.5) | 167 (43.5) | |
| **Marital Status** | | | | | |
| Single | 4.4 ± 1.3 | 0.703[†] | 182 (57.6) | 134 (42.4) | 1.341, 0.251[¥] |
| Married | 4.5 ± 1.3 | | 51 (51.0) | 49 (49.0) | |
| **Occupation** | | | | | |
| Unemployed | 4.5 ± 1.4 | 0.173[†] | 160 (53.7) | 138 (46.3) | 2.292, 0.154[¥] |
| Employed | 4.3 ± 1.1 | | 73 (61.9) | 45 (38.1) | |
| **Parity** | | | | | |
| 1 | 4.4 ± 1.3 | 0.512[†] | 173 (54.7) | 143 (45.3) | 0.851, 0.419[¥] |
| >1 | 4.4 ± 1.2 | | 60 (60.0) | 21.9 (40.0) | |
| **Education** | | | | | |
| None | 4.7 ± 1.1 | **0.031** | 10 (52.6) | 9 (47.4) | **12.7577, 0.005[‡]** |
| Primary | 4.2 ± 1.0[a] | | 40 (71.4) | 16 (28.6) | |
| JHS | 4.4 ± 1.3 | | 147 (57.6) | 108 (42.4) | |
| SHS | 4.7 ± 1.4[a] | | 36 (41.9) | 50 (58.1) | |
| **Income** | | | | | |
| No Income | 4.4 ± 1.3 | 0.092 | 182 (58.7) | 128 (41.3) | 5.100, 0.165[‡] |
| Below 100 | 4.4 ± 1.3 | | 22 (53.7) | 19 (46.3) | |
| Between 100–500 | 4.8 ± 1.3 | | 28 (45.9) | 33 (54.1) | |
| More than 500 | 5.0 ±0.8 | | 1 (25.0) | 3 (75.0) | |

Data are presented as frequency (percentage), mean ± SD (Standard deviation),

[‡] Chi-Square,

[¥] Fisher's exact test,

[†]- Mann Whitney test,

Kruskal-Wallis test, Bold values are significant at p<0.05. (a- mean difference between primary and SHS leaver participants, p = 0.042).

income status of the pregnant girls. The mean dietary diversity was higher (p = 0.008) among urban pregnant adolescents (4.6 ± 1.3) compared with rural adolescents (4.2 ± 1.2).

## Effect of household hunger and poverty, and adolescents' food aversion and cravings on dietary diversity

Table 4 presents the relationship between dietary diversity and household hunger scale, lived poverty, and eating behavior. HHS, LPI, and eating behavior did not vary significantly by dietary diversity (p>0.05). However, a large proportion of participants with low moderate LPI (61.7%, p = 0.607) and severe hunger (68.0%, p = 0.197) presented inadequate dietary diversity compared to the others. More participants who were practicing pica (62.5%), food aversion (63.8%), and food craving (57.5) had inadequate dietary diversity compared to those who did not (p>0.05). Pregnant adolescents with severe hunger (4.0 ± 1.2) had the least dietary diversity (p = 0.033) compared to moderate (4.1 ± 1.2) and no hunger for pregnant adolescents

**Table 4. Comparison of means and proportions of MDD-W by household hunger scale, lived poverty and eating behaviours.**

| VARIABLES | MEAN ± SD | P-VALUE | DIETARY DIVERSITY | | X², P-VALUE |
| --- | --- | --- | --- | --- | --- |
| | | | INADEQUATE | ADEQUATE | |
| **Household Hunger Scale** | | | | | |
| No Hunger | 4.5 ± 1.3 | **0.033** | 161 (53.1) | 142 (46.9) | 4.408, 0.110‡ |
| Moderate Hunger | 4.1 ± 1.2 | | 38 (60.3) | 25 (39.7) | |
| Severe Hunger | 4.0 ± 1.2 | | 34 (68.0) | 16 (32.0) | |
| **Lived Poverty Index** | | | | | |
| Low | 4.5 ± 1.3 | 0.261 | 107 (55.2) | 87 (44.8) | 1.813, 0.404‡ |
| Moderate | 4.6 ± 1.3 | | 38 (50.7) | 37 (49.3) | |
| High | 4.3 ± 1.3 | | 88 (59.9) | 59 (40.1) | |
| **Food Aversion** | | | | | |
| Yes | 4.3 ± 1.4 | **0.056†** | 109 (58.9) | 76 (41.1) | 1.144, 0.320¥ |
| No | 4.5 ± 1.2 | | 124 (53.7) | 107 (46.3) | |
| **Food Craving** | | | | | |
| Yes | 4.5 ± 1.3 | 0.223† | 143 (53.6) | 124 (46.4) | 0.818, 0.182¥ |
| No | 4.4 ± 1.3 | | 90 (60.4) | 59 (39.6) | |
| **Pica Practice** | | | | | |
| Yes | 4.4 ± 1.3 | 0.368† | 94 (58.7) | 66 (41.3) | 0.792, 0.417¥ |
| No | 4.5 ± 1.3 | | 139 (54.3) | 117 (45.7) | |

Data are presented as frequency (percentage),

‡ Chi-Square,

¥ Fisher's exact test,

† - Mann Whitney test,

Kruskal-Wallis test, DD- Diet diversity, P-value is significant at p<0.05.

(4.5 ± 1.3). Younger participants (13–15 years) had a higher mean score (4.7 ± 1.2) compared to older participants (aged 16–19 years) (4.3 ± 1.2) (p = 0.052). The mean dietary diversity (p = 0.056) was higher among participants who did not practice food aversion (4.5 ± 1.2) compared with those who practiced food aversion (4.3 ± 1.4). The mean dietary diversity did not significantly vary by poverty status (p = 0.261), pica practice (p = 0.368), and food craving (p = 0.223).

The results of between-subject effects, conducted in univariate and multivariate Generalized Linear Model are presented in Table 5. Among all the multiple variables tests of associations on dietary diversity, only hunger status (p = 0.028) and both food aversion and poverty status (p = 0.003) had a significant effect on the adolescents' dietary diversity.

## Predictors of dietary diversity

The predictors of inadequate dietary diversity of the pregnant adolescent are presented in Table 6. Rural dwellers had higher odds (AOR = 1.7, p = 0.035, 95%CI = 1.0–2.6) for inadequate DDS compared with urban adolescents. Pregnant adolescents with severe hunger had higher odds (AOR = 1.9, p = 0.053, 95% CI 1.1–3.8) for inadequate dietary diversity compared with those with no hunger. Younger adolescents (13–15 years age bracket) had higher odds (AOR = 1.2, p = 0.714, 95%CI = 0.5–2.5) compared with the older adolescents. Non-earning pregnant adolescents had higher odds (AOR = 1.3, p = 0.817, 95%CI = 0.2–10.2) for inadequate dietary diversity compared with those who earned between 500–100 cedis, those with

**Table 5. Tests of between-subject effects of multiple variables on dietary diversity.**

| Tests of Between-Subjects Effects | | | | | |
|---|---|---|---|---|---|
| **Dependent Variable: DD** | | | | | |
| **Source** | **Type III Sum of Squares** | **df** | **Mean Square** | **F** | **p-value** |
| Intercept | 625.879 | 1 | 625.879 | 389.414 | <0.001 |
| Community type | 0.261 | 1 | 0.261 | 0.162 | 0.687 |
| Age group | 0.184 | 1 | 0.184 | 0.115 | 0.735 |
| Marital status | 0.127 | 1 | 0.127 | 0.079 | 0.779 |
| Income status | 4.17 | 1 | 4.17 | 2.594 | 0.108 |
| Parity | 0.986 | 1 | 0.986 | 0.614 | 0.434 |
| Food aversion | 0.283 | 1 | 0.283 | 0.176 | 0.675 |
| Pica practice | 3.34 | 1 | 3.34 | 2.078 | 0.150 |
| Food craving | 0.662 | 1 | 0.662 | 0.412 | 0.521 |
| Lived Poverty Index | 0.164 | 2 | 0.082 | 0.051 | 0.950 |
| Household Hunger Scale | 11.611 | 2 | 5.805 | 3.612 | **0.028** |
| Lived Poverty Index * Household Hunger Scale | 4.854 | 4 | 1.213 | 0.755 | 0.555 |
| Food aversion * Lived Poverty Index | 18.624 | 2 | 9.312 | 5.794 | **0.003** |
| Food aversion * Household Hunger Scale | 0.462 | 2 | 0.231 | 0.144 | 0.866 |
| Error | 634.857 | 395 | 1.607 | | |

General Linear Model test for univariate and multivariate parameters, DD: Dietary Diversity of previous day dietary intake, determined from the FAO's Minimum Dietary Diversity–Women consisting of 10 food groups.

employment had higher odds (AOR = 1.3, p = 0.333, 95%CI = 0.8–2.0) compared with those with no employment. Pregnant adolescents who practiced food craving had higher odds (AOR = 1.3, p = 0.219, 95%CI = 0.8–2.0) for inadequate dietary diversity compared with those who did not practice food craving. Pregnant adolescents with low lived poverty had lower odds (OR = 0.7 p = 0.161, 95%CI = 0.4–1.2) for inadequate dietary diversity compared with those with high lived poverty.

## 4. Discussion

This study sought to assess dietary diversity and its association with socio-demographic factors and dietary factors among pregnant adolescents in rural and urban districts of Ghana. The mean DD score of our study participants was 4.4 less than the recommended score of 5 by the MDD-W. However, more than half of the participants (56%) did not consume the MDD-W score of at least 5 food groups. Again, this situation is slightly better for urban girls. The Ghana Micronutrient Survey in 2017 recorded a mean DDS of 4.40 among pregnant women of all ages [27]. This finding is similar to the adolescent girls in our study, suggesting that DDS for adult pregnant women and adolescents in Ghana are similar. Studies in the Northern part of Ghana [39] and Ashanti, Ghana [40] reported a mean DDS of 3.95 and 3.81 respectively which are slightly lower than our study. Studies in Bangladesh [29] and Kenya [41], respectively have reported higher WDDS of 5.1 and 5.6 among pregnant adolescents. Our study and the other studies both in and outside Ghana, all suggested dietary diversity among pregnant women, in general, is inadequate. For the adolescents, the extra nutrient needs for their growth as well as that of the foetus make these inadequacies much more worrying. Inadequate DDS also depicts that these pregnant adolescents are likely to have nutrients deficiencies, especially micronutrients, which could have a poor prognosis on birth outcome.

**Table 6. Predictors of inadequate dietary diversity among pregnant adolescents.**

| Variable | OR (95%CI) | p-value | AOR (95%CI) | p-value |
|---|---|---|---|---|
| | | Inadequate dietary diversity | | |
| **Community Type** | | | | |
| Rural | 1.7(1.1–2.5) | **0.009** | 1.7(1.0–2.6) | **0.035** |
| Urban | 1.0 | | 1.0 | |
| **Age Group (Years)** | | | | |
| 16–19 | 1.3(0.6–2.8) | 0.443 | 1.2(0.5–2.5) | 0.714 |
| 13–15 | 1.0 | | 1.0 | |
| **Education Group** | | | | |
| None | 0.5(0.2–1.4) | 0.208 | 0.6(0.2–1.8) | 0.360 |
| Primary | 0.8(0.4–1.6) | 0.592 | 0.9(0.4–1.9) | 0.802 |
| JHS | 0.9(0.6–1.6) | 0.836 | 1.0(0.6–1.7) | 0.994 |
| SHS | 1.0 | | 1.0 | |
| **Income Status (₵)** | | | | |
| None | 1.4(0.2–10.2) | 0.727 | 1.3(0.2–10.0) | 0.817 |
| Below 100 | 1.0(0.1–7.4) | 0.963 | 1.0(0.1–8.2) | 0.967 |
| Between 100–500 | 0.9(0.1–6.7) | 0.924 | 0.9(0.1–7.8) | 0.954 |
| Between 500–1000 | 1.0 | | 1.0 | |
| **Marital Status** | | | | |
| Single | 1.4(0.9–2.2) | 0.166 | 1.3(0.8–2.2) | 0.291 |
| Married | 1.0 | | 1.0 | |
| **Parity** | | | | |
| 1 pregnancy | 1.2(0.7–1.8) | 0.487 | 1.0(0.6–1.7) | 0.886 |
| >1 pregnancy | 1.0 | | 1.0 | |
| **Occupation** | | | | |
| Employed | 1.4(0.9–2.2) | 0.131 | 1.3(0.8–2.0) | 0.333 |
| Unemployed | 1.0 | | 1.0 | |
| **Food Aversion** | | | | |
| Yes | 0.9(0.6–1.3) | 0.472 | 0.8(0.5–1.2) | 0.258 |
| No | 1.0 | | 1.0 | |
| **Food Craving** | | | | |
| Yes | 1.3(0.8–1.9) | 0.261 | 1.3(0.8–2.0) | 0.219 |
| No | 1.0 | | 1.0 | |
| **Pica Practice** | | | | |
| Yes | 1.0(0.7–1.5) | 0.938 | 1.0(0.6–1.5) | 0.928 |
| No | 1.0 | | 1.0 | |
| **Lived Poverty Status** | | | | |
| Low | 0.8(0.5–1.4) | 0.433 | 0.7(0.4–1.2) | 0.161 |
| Moderate | 1.2(0.8–1.9) | 0.399 | 0.9(0.5–1.5) | 0.730 |
| High | 1.0 | | 1.0 | |
| **Household Hunger Status** | | | | |
| Moderate | 1.3(0.8–2.3) | 0.299 | 1.3(0.7–2.4) | 0.463 |
| Severe | 1.9(1.0–3.5) | **0.053** | 1.6(0.7–3.6) | 0.218 |
| No hunger | 1.0 | | 1.0 | |

Adjusted for Nutrition education given, OR- Odds ratio, AOR- Adjusted odds ratio, CI- Confidence interval.

## Determinants of inadequate dietary diversity

Among all the multiple variables tests of associations on dietary diversity, only hunger status (p = 0.028) and both food aversion and poverty status (p = 0.003) had a significant effect on the adolescents' dietary diversity. This suggests that food insecurity, food averting together with poverty during pregnancy might have contributed to the dietary diversity of studied adolescents. Also, the study found the significant predictors of inadequate dietary diversity were rural living and food insecurity during pregnancy. Rural dwellers had higher odds of inadequate DDS compared with urban adolescents. Contrary to [42] who found no significant difference among rural and urban residents (P = 0.067), our findings corroborate with [7,43] who reported that poor communities were at higher risk of having low DDS. This indicates that the poorer participants in the rural areas did not have adequate dietary diversity which was evident in more than half of their population being classified in the high LPI group. However, the urban girls may consume nutritious diets as these are readily available in urban centers, thus, the little available food could be inexpensive, nutritious, and diverse. Thus, urban areas could have improved access to diverse food more than rural areas.

Pregnant adolescents with severe hunger had higher odds of inadequate dietary diversity compared with those with no hunger. These results corroborate with [12] who reported 2 and 3 times increased odds for high DDS among pregnant women who ate three meals and above in a day and were food secured respectively. In our study, the HHS assessment found that more than a quarter of the participants experienced hunger at least once or twice in a month. A larger proportion of this percentage was from the rural participants with a corresponding greater inadequate DD as compared to their urban folks. The proportion of participants experiencing hunger in the rural areas (49%) might be explained by the level of poverty (70%), since a greater proportion of the rural participants (27%) who experienced severe hunger, also showed high LPI. Pregnant adolescents with low lived poverty index had lower odds for inadequate dietary diversity compared with those with high lived poverty. The mean LPI score for the study was 1.48, suggesting that the participants lived without each of the basic necessities once or twice over the past year before the study. The urban girls were again better. In a study to compare poverty status with the total intake of micronutrients in the USA, [44] found out that people living in poverty presented higher rates of inadequate nutrient intake. This also accords previous studies that age [45] and place of residence [46] are significant determinants of poverty in Kenya and Rwanda respectively. This suggests that pregnant adolescents living in poverty and deprived of food might be at risk of inadequate dietary intake.

## Socio-demographic factors and dietary diversity

Socio-demographic factors have been shown to determine dietary intake in many populations [47–49]. Our findings strongly showed that socio-demographic circumstances predicted DD of pregnant adolescents. Our study recruited more adolescents from urban areas. The minimum age of the participants was 13 years. Our findings showed that older adolescents (16–19 years age bracket) had higher odds for inadequate dietary diversity compared with younger adolescents. This may suggest adequate dietary diversity reduced with age, evident in the 0.5 increase in the mean MDD of younger girls (13–15 years). Non-earning pregnant adolescents had higher odds for inadequate dietary diversity compared with those who earned between 500–100 cedis, and those with employment had higher odds compared with those with no employment. Opposing to our findings, [7] reported three times increased odds for adequate DDS of participants who received income in Ethiopia. Our findings also showed that slightly more rural girls were employed and earned income compared with their urban counterparts. Meanwhile, the urban girls had a significantly better DDS (p = 0.008). It is likely that improved

access to quality diet is beyond income and employment but also availability that the food environment offers. The girls from the rural areas are likely to consume more monotonous meals from energy-dense foods (mainly starchy foods) they might have cultivated which is of a limited variety, compared to the urban girls where more varied food would be available to buy. In contrast, other studies [50,51] have reported that rural households with home garden and/ or livestock rearing often have higher DDS. Therefore, interventions that promote rural household's home gardening and poultry are recommended to boost the availability and consumption of diverse food.

### Eating behavior and dietary diversity

With regards to food groups, staples (99.3%) were the most consumed whilst dairy food sources (17%) were the least. Food groups such as eggs, pulses, nuts and seeds, dark green vegetables and other vitamin A-rich fruits and vegetables and other fruits were seldom consumed and from our studies, the diet of the majority is quite monotonous, energy-dense, and lack nutrients. Several studies have reported the over-dependence on cereal and root-based staples in Ghana [52–54]. Thus, their dietary patterns predicted an insufficient intake of micronutrients as well as fiber which explains the high rate of inadequate dietary diversity. Prevalence of food cravings in this study (64%) was similar to findings conducted in Ghana [55], Nigeria [56], and Iran [57] which were 67.7%, 61.3%, and 60% respectively but was slightly lower than findings from Kenya [58] and Ecuador [59] which were 74% and 69% respectively. Food aversion prevalence (45%) was comparable to findings which were also conducted in Ghana (44.8%) by [55] and in Kenya [58] but lower than those reported by [19] and [59] in Tanzania and Ecuador respectively. Current findings show the practice of pica during pregnancy (39%) was slightly lower than studies conducted in Ghana by [42,55,60] which were between 47–48% but higher than those conducted in Kenya [58] and Ethiopia [61] which were 27% and 30% respectively. Regression analysis showed that participants who practiced food craving had higher odds for inadequate DD. This finding may imply that the practice of craving food could lead to reduced dietary intake and consequently micronutrient inadequacy.

### Limitation

The use of the FAO's guidelines in measuring dietary diversity only takes into account the past 24-hour meal consumed by participants which might not reflect usual consumption habits. Also, it does not include the number of food items consumed as it only reflects economic access to food and not nutritional quality. Also, dietary intake can vary during lean or bumper seasons as well as during festivities which were not considered during this study. The rate of attendance to antenatal clinics by pregnant adolescents especially in the rural areas was very low basically due to stigmatization which led to a lesser number of participants being recruited from the rural areas. Besides the limitations, this study has provided new insight on factors that influence dietary diversity among pregnant adolescents to the scientific community especially in the Ghanaian context which is novel and interesting.

## 5. Conclusions

Dietary diversity among pregnant adolescents in the middle belt of Ghana is considered generally inadequate as over half of these adolescents had inadequate DDS. The consumption of less diversified foods which lacks micronutrients was high. The main predictors of inadequate dietary diversity among the pregnant adolescents used were being a rural dweller and experiencing severe hunger. Generally, the study predicted that rural dwelling pregnant adolescents were more likely to exhibit poverty, hunger, and consequently high inadequate dietary

diversity. Therefore, interventions to support and improve the socio-economic and livelihoods of pregnant adolescents are likely to improve dietary diversity and reduce the risk of malnutrition during pregnancy and adverse birth outcomes.

## Acknowledgments

We thank the directors of health and health workers in the health centers where the study took place, and most of all the pregnant adolescents who participated in the study.

## Author Contributions

**Conceptualization:** Reginald Adjetey Annan, Charles Apprey, Anthony Edusei, Wisdom Azanu, Herman Lutterodt.

**Data curation:** Linda Afriyie Gyimah, Linda Nana Esi Aduku, Odeafo Asamoah-Boakye.

**Formal analysis:** Linda Afriyie Gyimah, Reginald Adjetey Annan, Odeafo Asamoah-Boakye.

**Funding acquisition:** Reginald Adjetey Annan, Charles Apprey.

**Investigation:** Linda Afriyie Gyimah, Linda Nana Esi Aduku.

**Methodology:** Reginald Adjetey Annan, Charles Apprey.

**Project administration:** Reginald Adjetey Annan, Charles Apprey.

**Supervision:** Reginald Adjetey Annan, Charles Apprey, Anthony Edusei.

**Validation:** Reginald Adjetey Annan.

**Visualization:** Linda Afriyie Gyimah.

**Writing – original draft:** Linda Afriyie Gyimah.

**Writing – review & editing:** Reginald Adjetey Annan, Charles Apprey, Anthony Edusei, Wisdom Azanu.

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
