## [Decision Letter · Decision Letter 0]

15 Oct 2020

PONE-D-20-16523

Dietary diversity and its correlates among pregnant adolescent girls in Ghana

PLOS ONE

Dear Dr. Gyimah,

Thank you for submitting your manuscript to PLOS ONE. After careful consideration, we feel that it has merit but does not fully meet PLOS ONE’s publication criteria as it currently stands. Therefore, we invite you to submit a revised version of the manuscript that addresses the points raised during the review process.

Please submit your revised manuscript by  December 14, 2020. If you will need more time than this to complete your revisions, please reply to this message or contact the journal office at plosone@plos.org. Please include the following items when submitting your revised manuscript:

We look forward to receiving your revised manuscript.

Kind regards,

Mohammad Rifat Haider, MBBS, MHE, MPS, PhD

Academic Editor

PLOS ONE

Journal Requirements:

2. Please amend either the abstract on the online submission form (via Edit Submission) or the abstract in the manuscript so that they are identical.

Additional Editor Comments (if provided):

Please take care of the reviewers' comments. Also make sure the methods are clearly mentioned and discussion is based on the findings.

Reviewers' comments:

Reviewer's Responses to Questions

**Comments to the Author**

1. Is the manuscript technically sound, and do the data support the conclusions?

Reviewer #1: Partly

Reviewer #2: Yes

2. Has the statistical analysis been performed appropriately and rigorously? 

Reviewer #1: Yes

Reviewer #2: No

3. Have the authors made all data underlying the findings in their manuscript fully available?

Reviewer #1: Yes

Reviewer #2: Yes

4. Is the manuscript presented in an intelligible fashion and written in standard English?

Reviewer #1: Yes

Reviewer #2: Yes

5. Review Comments to the Author

Reviewer #1: The paper titled “Dietary diversity and its correlates among pregnant adolescent girls in Ghana” is an interesting piece of work and has the potential to provide some evidence for appropriate interventions to address inadequate dietary intake among pregnant adolescents in Ghana. Notwithstanding, the manuscript has a number of shortfalls that need to be addressed to strengthen it. The details of my review is presented below:

Introduction:

1. This is largely well presented. However, the correlates (e.g. various socio-demographic factors, including cultural issues) and how they influence (negatively or positively) adolescents dietary diversity did not come out strongly. The authors should take this into account when revising the manuscript.

2. I will also suggest that the authors should strengthen the last paragraph a bit to convince the reader why the study essential at this time.

Methods:

1. The methods section will need a bit more work to make it very clear and robust. For example, the study design (larger study) needs to be explained in a bit more detail. As it stands, it is not clear what went into the design

2. It is not also clear the component of the larger study that was used for the present study

3. The description of how the sample size was determined or calculated is not also clear. E.g. It may not be enough to use only two parameters to compute or determine a sample size. If the sample size was not statistically determined, the authors could explain why? And if it were, a brief description of the estimation process would be helpful. The robustness of study design and sampling is critical to the scientific validity of any study findings

4. What informed the researchers’ decision to use the prevalence of birth weight instead of other nutrition indicators

5. The authors should explain how the health facilities were selected, and why? E.g. were they randomly selected?

6. Why was every pregnant adolescent who visited the health facilities recruited without using any form of randomisation?

7. The authors did not address ethical issues relating to the study. This must be done under a sub-heading.

Data collection:

1. Who were the data collectors and how were they trained

2. How was the longitudinal data collected, at what interval and for how long

3. Listing the variables under this section is inappropriate. The section should focus on the data collection processes

4. A separate subsections focusing on the various variables should be created, e.g.:

a. Dependent variables—define or describe how they were computed (if necessary) and treated in the analysis.

b. And another subsection on independent variables. Also, define or explain how they were computed (if applicable) and categorised/treated

5. Why did the researchers use WDDS instead of the most recent MDD-W indicator? Please, clarify?

6. The authors indicated that the dietary diversity score was made up 10 food groups, yet the WDDS use 9 food groups. It is rather the MDD-W that has 10 food groups? The author should clarify this inconsistency.

7. The authors also categorised the WDDS as follows per FAO guidelines: low (<=3), medium (4-5 food groups) and high (>=6 food groups).

Data analysis:

1. Although the authors classified the WDDS into 3 categories as can be seen above, when it came to the analysis, the WDDS was dichotomised. Why is that the case?

2. The authors could have used multinomial logistic regression to analyse the three categories instead of dichotomising for the purpose of using binomial logistic regression.

3. Any clarification of why they chose binomial logistic regression will be helpful.

4. The authors indicated that the present study is part of a larger longitudinal study. However, it is unclear the component of the larger study the data used for this analysis came from. E.g. baseline etc. or this was a cross-sectional survey nested within the longitudinal study?

5. List the independent variables that where statistical significant and the level of significance used to determine this?

6. How was the predictors entered into the regression model?

Results:

1. The authors have given so much attention to the descriptive results, which to a large extent, dwarfed the main analysis of the paper (Table 6).

2. If some of the descriptive tables can be merged and the text shortened, it would be helpful to the reader.

3. It is also confusing how some of the results are presented in the tables. E.g. in Tables 3 and 4, the authors presented RR and CIs for some of the variables and none for others.

4. Also, presenting RR & CIs in this type of analysis appears a bit problematic

5. The authors should take a second look at how they presented Table 6 results in the text. E.g. urban dwellers (OR=1.7, p=0.034, 95%CI=1.0-2.8) had higher odds for adequate DDS than rural----. The results in parenthesis should come directly after higher odds. This should be correct throughout this section

6. Further, the way the authors interpreted the ORs is also not appropriate; the use of the word “than” in comparing the variable of interest to the reference group is problematic. E.g. if you say females have high odds of xxx than males, it would be expected that you will provide the values/results for the males so that the reader can see the difference, but we don’t usually present reference group results

--Therefore, in interpreting ORs, words such as compare with/to, relative to etc. are used. The authors should revise this section to reflect the right interpretation of ORs. This section is the core of the paper, and I, therefore, expect the authors to pay more attention to it

7. The authors did not use adjusted odds ratio. Any reason why this is the case?

The results section is difficult to follow. I will therefore advice the authors to do a bit more work on this section to make it flow more coherent.

Discussion:

1. This section also needs to be reworked on. The authors spent a lot of time discussing the descriptive results, which do not contribute much to the scientific robustness of the paper. We hardly discuss descriptive results in scientific papers. I will suggest this should be cut out.

2. The regression results are the heart of the paper and should be the focus of the discussion section. I will suggest that the authors’ focus on discussing the exciting findings obtained from the regression modelling in more detailed, and in the context of the existing literature. This will enhance the scientific robustness of the work.

3. The discussion should focus on the most striking findings and not all statistically significant findings

4. The authors should also avoid including the p-values, OR and CIs in this section. Those have already been captured in the results section.

Limitations:

1. Does the study have missing data issues? If yes, please, could you discuss these and indicate how the missingness was handled in the analysis.

2. What about the issue of external validity or generalizability of the findings?

Conclusions:

1. The conclusions are quite weak. The section should be strengthened using the most striking findings from the regression results.

Reviewer #2: General Comments: This cross-sectional study based on longitudinal data assessed dietary diversity and its correlates in adolescent pregnant girls in Ashanti Region, Ghana. Although the manuscript is generally well written, it could benefit from revision.

Revisions being requested

Abstract

1. Although a 3-day repeated 24-hour dietary recall data were collected, the results reported in the manuscript are based on only the first recalls according to your methods. Therefore it is misleading to report that the analyses presented in the manuscript were based on a 3-day 24-hour recalls.

2. Report the mean age of the girls.

3. The odds ratios were estimated in a multivariate logistic regression model so should be reported as adjusted.

Introduction

4. You report that dietary diversity and its correlates in pregnant adolescent girls have not been studied in Ghana, but what have studies conducted elsewhere on pregnant adolescent girls found?

5. Paragraph 4, line 6: Correct grammar in the sentence: “Pica, another form of craving involves …”.

6. Paragraph 4, line 8: Correct grammar in the sentence: “It is useful in protecting against pathogens …”

7. Your study is premised on the fact that improving dietary diversity will contribute to desirable pregnancy outcomes so it will be good to provide some statistics to back this.

8. The last sentence “This study assessed …” should include “Region” after “Ashanti”.

Methods

Study setting

9. First paragraph, line 4: Revise the sentence “Each district has a …” to “Each district has a district hospital, health centres and several Community-based Health Planning Services (CHPS) compounds.”

Study design

10. First paragraph, line 2: The sentence “The study involves 416 pregnant adolescents with gestational age up to 32 gestational weeks …” should be revised to “The study involves 416 adolescents with gestational age up to 32 weeks …”

Data collection

11. Line 1: The sentence “A standardized questionnaire were …” should be revised to “A standardized questionnaire was … ”

12. Line 3: The sentence “Data on age and parity …” should end with “books” and not “book”.

13. The last sentence is not complete.

Assessment of dietary diversity and eating behaviour

14. Line 3: The sentence “The FAO’s Women’s Dietary Diversity (WDDS) …” should be “The FAO’s Women’s Dietary Diversity Score (WDDS) …”

15. Line 7: Correct grammar in the sentence: “The first 24-hour recall …”

16. It is not clear how the WDDS was calculated, please explain the scoring used.

17. The adequate category (5-9) of WDDS should include a score of 10.

Assessment of food availability and poverty

18. Line 3: Correct grammar in the sentence “Scores between <2 means …”

19. Last sentence: Check the categories of lived poverty index scores.

Data analysis

20. The sentence “Data were first entered into Microsoft Excel 2019 and thoroughly cleaned to eliminate errors in the data.” should be revised as “Data were first entered into Microsoft Excel 2019 and thoroughly cleaned to eliminate errors.”

21. Line 3: Start the sentence with a capital letter (S).

22. Line 5: The sentence “Descriptive statistics was …” should be revised to “Descriptive statistics were …”

23. Line 10: The sentence “A non-parametric …” should be “Non-parametric tests … were ….”

Results

24. Paragraph 1, line 3: I do not think the sentence “More rural than urban participants were younger teenagers (8.1% versus 7.4%, p=0.853) …” is accurate. This is because although 8.1% is more than 7.4%, the difference in the rates can be explained by chance given the insignificant p-value. The same applies to paragraph, line 4: Sentence “More participants who were practising …”

25. Table 1: The category of income “Between 500” should be “Between 100 and 500”.

26. Table 3: Please specify in the data analysis section why Fisher’s exact test was used.

27. Table 3: While I agree with the use of Chi-square test for education and Fisher’s exact test for income, it is not clear to me why Fisher’s exact test was used for community type, age group, marital status, occupation, and parity.

28. Table 4: The variable “Low Poverty Index” should be corrected to “Livid Poverty Index”.

29. The heading “P VALUE” should be “X2, P VALUE”.

30. Table 4: It is not clear why Fisher’s exact test was used for food aversion, food craving and pica practice.

31. Table 6: Please use adjusted odds ratio instead of odds ratio.

32. Table 6: Marital status is missing from this table.

33. Table 6: The 95% confidence interval, 1.1-136.5, for the category of income “More than 500” is too wide. You can consider merging the categories “Between 100-500” and “More than 500” to rectify this.

34. Why did you calculate Relative Risks for bivariate analyses (Tables 3 and 4) and Odds Ratios for multivariate analysis (Table 6) for the same set of variables? For comparison, I will prefer this same measure of association at both levels.

Discussion

35. First sentence: I think you studied dietary factors (food aversion, food craving and pica practice) in addition to socio-demographic factors.

36. Paragraph 2, line 2: In the sentence “More than half …” I do not think the DDS was for the last 3 days.

37. Paragraph 2, line 6: “A study in …” should be “Studies in …”

38. Paragraph 2, line 9: Revise the sentence “Our study, the others in Ghana, and others compared outside Ghana, …”

39. Paragraph 3, line 4: Revise the sentence “A possible explanation is that improve …”

6. PLOS authors have the option to publish the peer review history of their article (what does this mean?). If published, this will include your full peer review and any attached files.

Reviewer #1: **Yes: **Dr Dickson A Amugsi

Reviewer #2: No

---

## [Author Response · Author response to Decision Letter 0]

10 Dec 2020

I have uploaded the response to the rebuttal letter from both reviewers

---

## [Decision Letter · Decision Letter 1]

18 Feb 2021

Dietary diversity and its correlates among pregnant adolescent girls in Ghana

PONE-D-20-16523R1

Dear Dr. Gyimah,

We’re pleased to inform you that your manuscript has been judged scientifically suitable for publication and will be formally accepted for publication once it meets all outstanding technical requirements.

Kind regards,

Mohammad Rifat Haider, MBBS, MHE, MPS, PhD

Academic Editor

PLOS ONE

Additional Editor Comments (optional):

Reviewers' comments:

Reviewer's Responses to Questions

**Comments to the Author**

1. If the authors have adequately addressed your comments raised in a previous round of review and you feel that this manuscript is now acceptable for publication, you may indicate that here to bypass the “Comments to the Author” section, enter your conflict of interest statement in the “Confidential to Editor” section, and submit your "Accept" recommendation.

Reviewer #2: All comments have been addressed

2. Is the manuscript technically sound, and do the data support the conclusions?

Reviewer #2: Yes

3. Has the statistical analysis been performed appropriately and rigorously? 

Reviewer #2: Yes

4. Have the authors made all data underlying the findings in their manuscript fully available?

Reviewer #2: Yes

5. Is the manuscript presented in an intelligible fashion and written in standard English?

Reviewer #2: Yes

6. Review Comments to the Author

Reviewer #2: (No Response)

7. PLOS authors have the option to publish the peer review history of their article (what does this mean?). If published, this will include your full peer review and any attached files.

Reviewer #2: No

---

## [Editor Report · Acceptance letter]

24 Feb 2021

PONE-D-20-16523R1 

Dietary diversity and its correlates among pregnant adolescent girls in Ghana 

Dear Dr. Gyimah:

I'm pleased to inform you that your manuscript has been deemed suitable for publication in PLOS ONE. Congratulations! Your manuscript is now with our production department. 

Kind regards, 

on behalf of

Dr. Mohammad Rifat Haider 

Academic Editor

PLOS ONE